# Human and nonhuman primate meninges harbor lymphatic vessels that can be visualized noninvasively by MRI

Martina Absinta[1†], Seung-Kwon Ha[1†], Govind Nair[1], Pascal Sati[1], Nicholas J Luciano[1], Maryknoll Palisoc[2], Antoine Louveau[3], Kareem A Zaghloul[4], Stefania Pittaluga[2], Jonathan Kipnis[3], Daniel S Reich[1*]

[1]Translational Neuroradiology Section, National Institute of Neurological Disorders and Stroke, National Institutes of Health, Bethesda, United States; [2]Hematopathology Section, Laboratory of Pathology, National Cancer Institute, National Institutes of Health, Bethesda, United States; [3]Center for Brain Immunology and Glia, Department of Neuroscience, School of Medicine, University of Virginia, Charlottesville, United States; [4]Surgical Neurology Branch, National Institute of Neurological Disorders and Stroke, National Institutes of Health, Bethesda, United States

**\*For correspondence:**
reichds@ninds.nih.gov

[†]These authors contributed equally to this work

**Abstract** Here, we report the existence of meningeal lymphatic vessels in human and nonhuman primates (common marmoset monkeys) and the feasibility of noninvasively imaging and mapping them in vivo with high-resolution, clinical MRI. On T2-FLAIR and T1-weighted black-blood imaging, lymphatic vessels enhance with gadobutrol, a gadolinium-based contrast agent with high propensity to extravasate across a permeable capillary endothelial barrier, but not with gadofosveset, a blood-pool contrast agent. The topography of these vessels, running alongside dural venous sinuses, recapitulates the meningeal lymphatic system of rodents. In primates, meningeal lymphatics display a typical panel of lymphatic endothelial markers by immunohistochemistry. This discovery holds promise for better understanding the normal physiology of lymphatic drainage from the central nervous system and potential aberrations in neurological diseases.

DOI: https://doi.org/10.7554/eLife.29738.001

## Introduction

Recent reports (*Aspelund et al., 2015*; *Louveau et al., 2015b*) described the existence of a network of true lymphatic vessels within the mammalian dura mater that runs alongside blood vessels, notably the superior sagittal and transverse sinuses. The dural lymphatic vessels display typical immunohistochemical markers that identify lymphatic vessels elsewhere in the body. They provide an alternate conduit for drainage of immune cells and cerebrospinal fluid (CSF) from the brain, beyond previously described pathways of flow: via arachnoid granulations into the dural venous sinuses, and via the cribriform plate into the ethmoid region (*Weller et al., 2009*). Although early reports, based on injections of India ink into the cisterna magna of the rat, suggested that the dural pathway accounts for only a minority of the drainage (*Kida et al., 1993*), the more recent studies (*Aspelund et al., 2015*; *Louveau et al., 2015b*), which are based on injections of fluorescent tracers and in vivo microscopy, indicate that the dural system may be substantially more important for drainage of macromolecules and immune cells than previously realized.

Whether a similar network of dural lymphatics is present in primates remains unknown. Moreover, noninvasive visualization of the dural lymphatics – a necessary first step to understanding their

**eLife digest** How does the brain rid itself of waste products? Other organs in the body achieve this via a system called the lymphatic system. A network of lymphatic vessels extends throughout the body in a pattern similar to that of blood vessels. Waste products from cells, plus bacteria, viruses and excess fluids drain out of the body's tissues into lymphatic vessels, which transfer them to the bloodstream. Blood vessels then carry the waste products to the kidneys, which filter them out for excretion. Lymphatic vessels are also a highway for circulation of white blood cells, which fight infections, and are therefore an important part of the immune system.

Unlike other organs, the brain does not contain lymphatic vessels. So how does it remove waste? Some of the brain's waste products enter the fluid that bathes and protects the brain – the cerebrospinal fluid – before being disposed of via the bloodstream. However, recent studies in rodents have also shown the presence of lymphatic vessels inside the outer membrane surrounding the brain, the dura mater.

Absinta, Ha et al. now show that the dura mater of people and marmoset monkeys contains lymphatic vessels too. Spotting lymphatic vessels is challenging because they resemble blood vessels, which are much more numerous. In addition, Absinta, Ha et al. found a way to visualize the lymphatic vessels in the dura mater using brain magnetic resonance imaging, and could confirm that lymphatic vessels are present in autopsy tissue using special staining methods.

For magnetic resonance imaging, monkeys and human volunteers received an injection of a dye-like substance called gadolinium, which travels via the bloodstream to the brain. In the dura mater, gadolinium leaks out of blood vessels and collects inside lymphatic vessels, which show up as bright white areas on brain scans. To confirm that the white areas were lymphatic vessels, the experiment was repeated using a different dye that does not leak out of blood vessels. As expected, the signals observed in the previous brain scans did not appear.

By visualizing the lymphatic system, this technique makes it possible to study how the brain removes waste products and circulates white blood cells, and to examine whether this process is impaired in aging or disease.

DOI: https://doi.org/10.7554/eLife.29738.002

normal physiology and potential aberrations in neurological diseases – has not been reported. We therefore verified pathologically the existence of a dural lymphatic network in human and nonhuman primates (common marmoset monkeys) and evaluated two magnetic resonance imaging (MRI) techniques that might enable its visualization in vivo. First, the T2-weighted fluid-attenuation inversion recovery (T2-FLAIR) pulse sequence, which is the clinical standard for detecting lesions within the brain parenchyma, is highly sensitive to the presence of gadolinium-based contrast agents in the CSF (*Mamourian et al., 2000*; *Mathews et al., 1999*; *Absinta et al., 2015*). Second, 'black-blood' imaging sequences, which are typically used for measurement of vascular wall thickness or detection of atherosclerotic plaque, are tuned to darken the contents of blood vessels (even when they contain a gadolinium-based contrast agent), but in the process the images may highlight vessels with other contents and flow properties (*Mandell et al., 2017*). For comparison, we also acquired a postcontrast T1-weighted Magnetization Prepared Rapid Acquisition of Gradient Echoes (MPRAGE) MRI sequence, which is widely implemented for structural brain imaging and depicts avid enhancement of dura mater and blood vessels, but which would not be expected to discriminate lymphatic vessels.

## Results and discussion

Cerebral blood vessels have a highly regulated blood-brain barrier, protecting the neuropil from many contents of the circulating blood. Under physiological conditions, the blood-brain barrier prevents gadolinium-based chelates in standard clinical use from passing into the Virchow-Robin perivascular spaces and parenchyma, so that these structures do not enhance on MRI. On the other hand, dural blood vessels lack a blood-meningeal barrier, enabling leakage of circulating fluids and small substances, including gadolinium-based compounds. This explains the thin, though often incomplete, dural enhancement that is seen on conventional T1-weighted MRI scans under

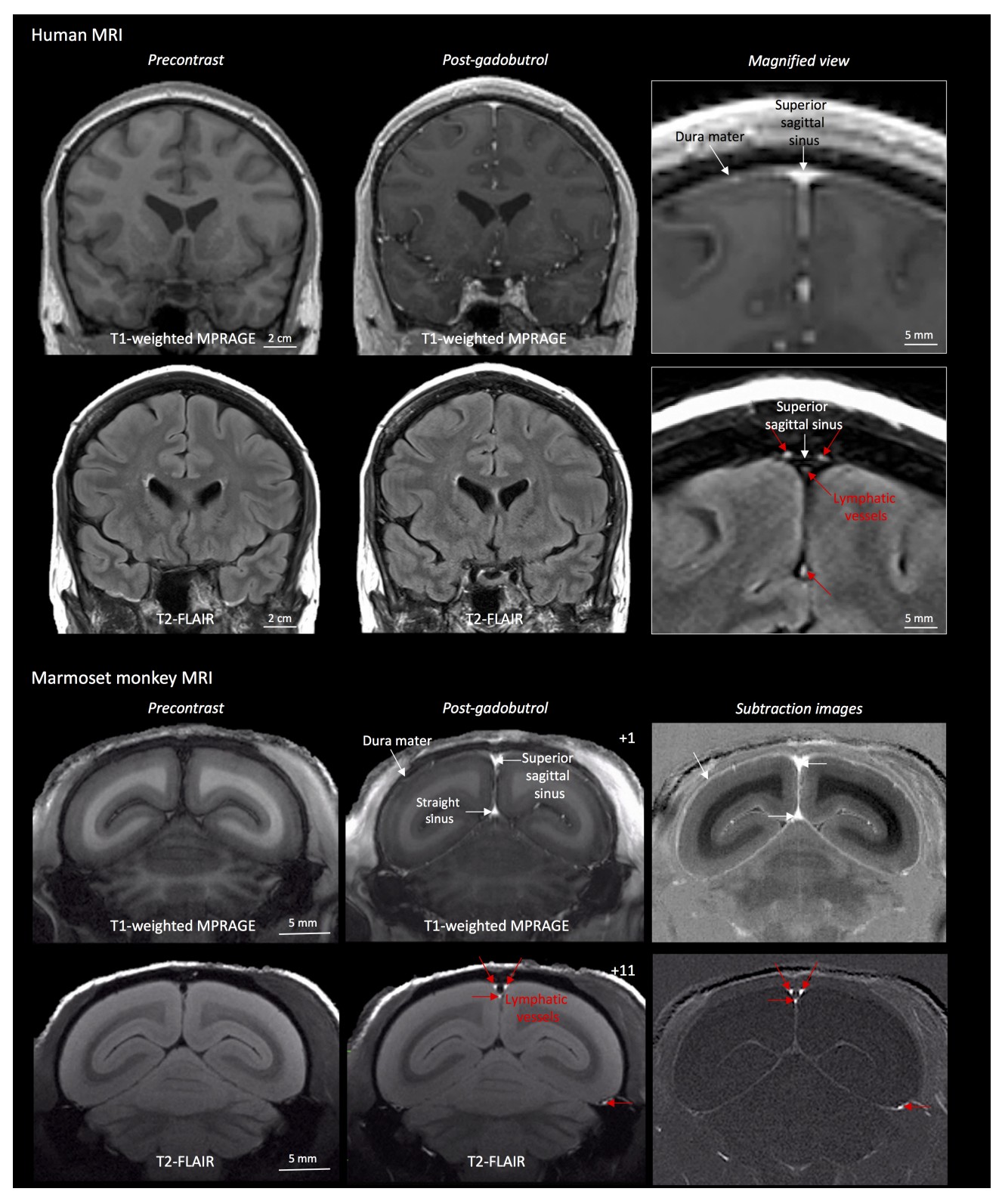

**Figure 1.** MRI-visualization of dural lymphatic vessels in human and nonhuman primates. In both species, conventional post-gadobutrol coronal T1-weighted MRI is unable to discriminate lymphatic vessels due to diffuse physiological enhancement of the dura (arrows) and blood vessels, including the superior sagittal sinus and straight sinus (arrows). On post-gadobutrol coronal T2-FLAIR and subtraction images, the dura does not enhance, and

*Figure 1 continued on next page*

*Figure 1 continued*

lymphatic vessels (red arrows), running alongside the venous dural sinuses and within the falx cerebri, can be appreciated. Numbers refer to minutes after the intravenous administration of gadobutrol.

DOI: https://doi.org/10.7554/eLife.29738.003

The following figure supplement is available for figure 1:

**Figure supplement 1.** 3D rendering of human dural lymphatics.

DOI: https://doi.org/10.7554/eLife.29738.004

physiological conditions (*Figure 1*), as well as its abnormal diffuse or localized thickening in a variety of pathological conditions (*Smirniotopoulos et al., 2007*; *Antony et al., 2015*). Using high-resolution (in-plane resolution 270 × 270 μm or finer) T2-FLAIR and T1-weighted black-blood MRI images, obtained after the intravenous injection of standard FDA-approved contrast material (gadobutrol), we were able to visualize the collection of interstitial gadolinium within dural lymphatic vessels (maximum apparent diameter ~1 mm) in 5/5 human healthy volunteers and 3/3 common marmoset monkeys (*Figure 1*). Our results suggest that in the dura, similar to many other organs throughout the body, small intravascular molecules extravasate into the interstitium and then, under a hydrostatic pressure gradient, collect into lymphatic capillaries through a loose lymphatic endothelium (*Sharma et al., 2008*).

To further test this hypothesis, meningeal lymphatics were also assessed using a second gadolinium-based contrast agent, gadofosveset, a blood-pool contrast agent suitable for angiography (*Lauffer et al., 1998*). Gadofosveset binds reversibly to serum albumin, increasing its molecular weight from 0.9 to 67 kDa. Under physiological conditions, albumin has a low transcapillary exchange rate into the interstitial compartment, estimated to be on the order of 5% per hour, which explains the propensity of gadofosveset to remain within blood vessels (*Richardson et al., 2015*). In both species, gadofosveset did not reveal dural lymphatics, especially on T1-black blood images (*Figure 2* and *Figure 2—figure supplement 1*). As expected, on T1-weighted MPRAGE images, gadofosveset provided superior intravascular enhancement, in both meningeal and parenchymal blood vessels, compared to gadobutrol (*Figure 2*).

On 3D-rendering of subtraction MRI images (*Videos 1–2*, *Figure 1—figure supplement 1*), dural lymphatics are seen running parallel to the dural venous sinuses, especially the superior sagittal and straight sinuses, and alongside branches of the middle meningeal artery. The topography of the meningeal lymphatics fits with the previously described network in rodents as well as our neuropathological data (*Figures 3* and *4*). It is worth noting that the lymphatics visualized by MRI are large slow-flow lymphatic ducts, whereas blind-ending and small lymphatic capillaries, clearly seen by histopathology (*Figure 3* and *Figure 3—figure supplement 1*), are unlikely to be revealed by MRI. The induction of experimental autoimmune encephalomyelitis (EAE) did not affect detection of dural lymphatic vessel in either of the two animals that we tested (not shown).

To support our in vivo imaging results, we further investigated the existence and topography of lymphatics in coronal and longitudinal sections of human and marmoset dura mater. To accomplish this, we tested a variety of putative lymphatic endothelial markers and found that selective double immunostaining for D2-40 podoplanin/CD31 and for PROX1/CD31 was the most effective strategy in discriminating lymphatic *vs.* venous blood vessels in dura samples – a challenging task since lymphatics sprout from transdifferentiation of venous endothelium (*Ny et al., 2005*; *Yaniv et al., 2006*; *Srinivasan et al., 2007*; *Aspelund et al., 2014*; *Lowe et al., 2015*) and persistently share some endothelial markers. A branched network of lymphatics was clearly seen within the dura mater. On D2-40 podoplanin/CD31 double staining, we identified a total of 93 human dural lymphatics; most were collapsed, explaining the large range of maximum transverse diameters (range = 7–842 μm, mean = 125 μm, standard deviation = 161 μm). The density of dural lymphatics was higher around the venous sinuses than in more lateral areas of the dura, and higher within the meningeal layer than the periosteal layer of the dura. As expected, red blood cells were not seen within lymphatics. In marmosets, direct comparison between in vivo MRI and histopathology was performed (*Absinta et al., 2014*; *Guy et al., 2016*; *Luciano et al., 2016*). As shown in *Figure 4* and *Figure 4—figure supplement 1*, the three dural vessels detected on coronal postcontrast T2-FLAIR and on subtraction images colocalize with three clusters of dural cells expressing the full panel of lymphatic

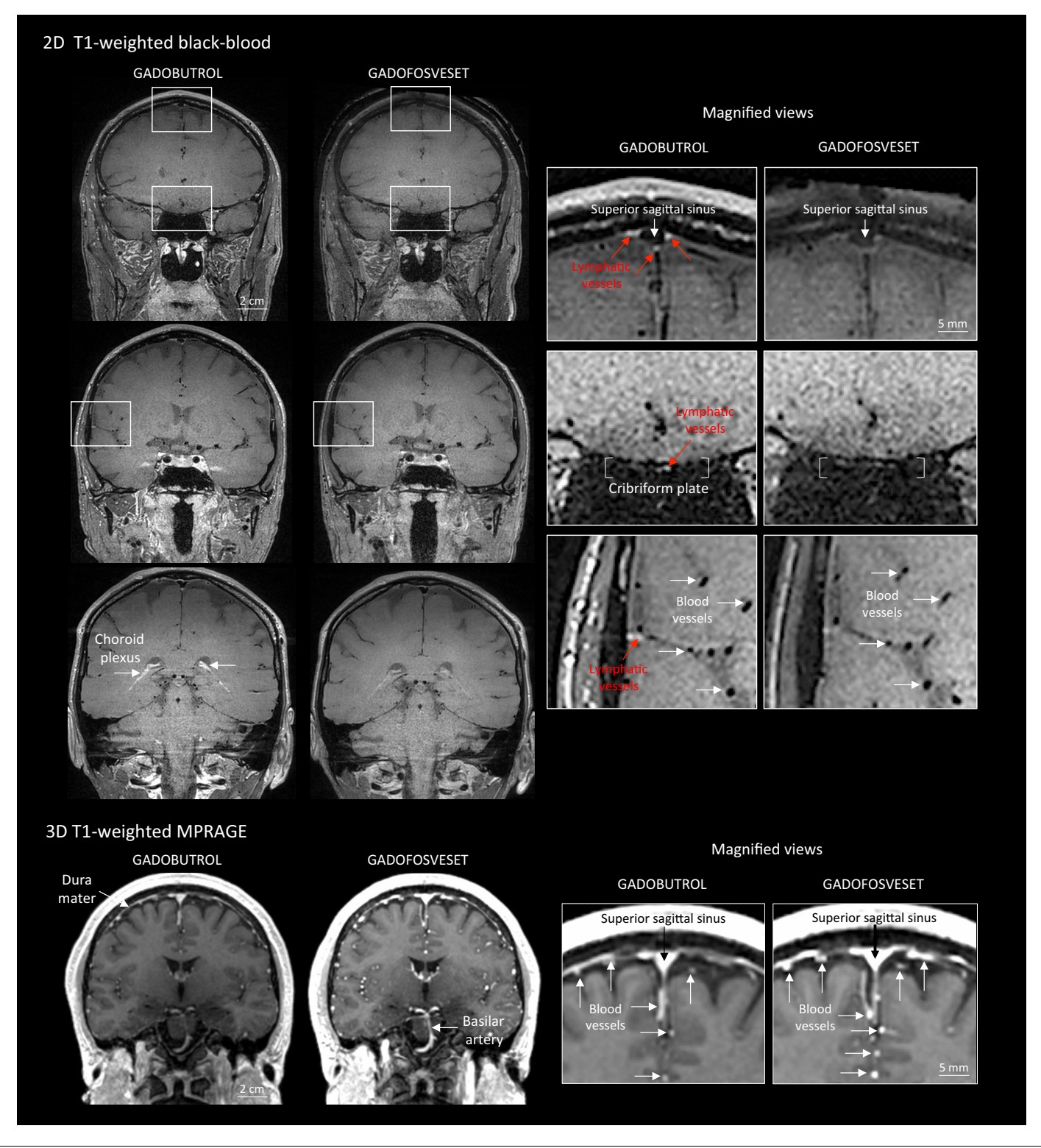

**Figure 2.** Gadobutrol vs. gadofosveset in MRI-visualization of dural lymphatic vessels. Coronal T1-weighted black-blood images were acquired after intravenous injection of two different gadolinium-based contrast agents (31 min after gadobutrol and 42 min after gadofosveset), during two MRI sessions separated by one week. Dural lymphatics (red arrows in magnified view boxes) were better discerned using gadobutrol (standard MRI contrast agent, which readily enters the dura) compared to gadofosveset (serum albumin-binding contrast agent, which remains largely intravascular) and were localized around dural sinuses, middle meningeal artery, and cribriform plate (white arrows). Notably, the choroid plexus (white arrows) enhanced less

*Figure 2 continued on next page*

*Figure 2 continued*

with gadofosveset than gadobutrol, whereas meningeal and parenchymal blood vessels (both veins and arteries) did not enhance with any contrast agent and appeared black. On conventional T1-weighted MPRAGE images, meningeal and parenchymal blood vessels enhanced with both contrast agents, more clearly with gadofosveset.

DOI: https://doi.org/10.7554/eLife.29738.005

The following figure supplement is available for figure 2:

**Figure supplement 1.** MRI visualization of dural lymphatic vessels in a healthy marmoset with different contrast agents.

DOI: https://doi.org/10.7554/eLife.29738.006

endothelial markers (LYVE-1, D2-40 podoplanin, PROX1, COUP-TFII) and CCL21, a chemokine implicated in lymphatic transmigration.

In the ongoing debate about the precise localization of lymphatics within the meninges (either completely within the dura or 'shared' between the dura and the arachnoid) (*Kipnis, 2016*; *Raper et al., 2016*), our pathological data clearly show that at least some lymphatics are contained entirely within the dura (*Figures 3* and *4*). On limited evaluation, we were unable to visualize lymphatics within the leptomeninges, but additional dedicated studies, ideally with nonconventional tissue-preparation methods, are warranted to fully explore this possibility. A comprehensive map of the meningeal lymphatic network would have implications for unraveling the ways in which the meningeal lymphatics participate in waste clearance and in immune cell trafficking within the central nervous system (*Louveau et al., 2015a*; *Kipnis, 2016*; *Raper et al., 2016*).

In inflammatory pathological conditions, cellular migration toward the dural lymphatics might be profoundly enhanced by specific signaling and lymphatic plasticity (*Alitalo et al., 2005*; *Kim et al., 2012*; *Stacker et al., 2014*). Noteworthy in this context are clusters of extravascular CD3+ lymphocytes and CD68+ phagocytic meningeal macrophages that we observed in the dura of several multiple sclerosis autopsies (not shown), confirming intense immune cell trafficking and communication. Indeed, without proper normative comparison, we cannot rule out that the extensive presence of small lymphatics observed in multiple sclerosis dura samples is the result of inflammation-mediated lymphoangiogenesis. On the other hand, lymphatic dysfunction might impair waste clearance in neurodegenerative diseases and aging, in line with the recently captured deposition of β-amyloid in human dura in elderly people (*Kovacs et al., 2016*).

Differently from experiments implementing injections of tracers within brain structures, here we aimed primarily to image dural lymphatic vessels in human and nonhuman primates, but we could not prove whether dural lymphatic vessels drain immune cells, CSF, or other substances from the brain to deep cervical lymph nodes, nor could we assess any link with the glymphatic system (*Iliff et al., 2012*; *Xie et al., 2013*; *Iliff et al., 2013*). Such an analysis would probably require injection of specific MRI-detectable tracers and acquisition of MRI time-series (not thoroughly explored in the current work), an important future research direction for the nonhuman primate work.

Overall, our data clearly and consistently demonstrate the existence of lymphatic vessels within the dura mater of human and nonhuman primates. Together with recent studies in rodents, our results show that the meningeal lymphatic system is evolutionarily conserved in mammals and confirm, after exactly two centuries, what the Italian anatomist Paolo Mascagni speculated were lymphatic vessels at the surface of human brain (*Mascagni and Bellini, 1816*).

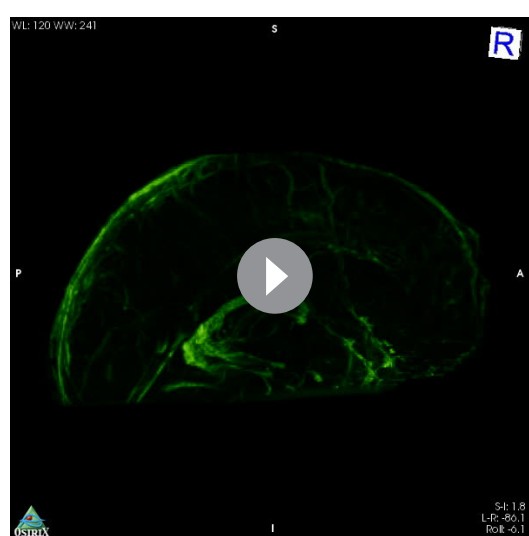

**Video 1.** 3D-rendering of dural lymphatics (green) in a 47 year old woman from skull-stripped subtraction T1-black-blood images (horizontal view, 180 degrees, 7 frames/minute).

DOI: https://doi.org/10.7554/eLife.29738.014

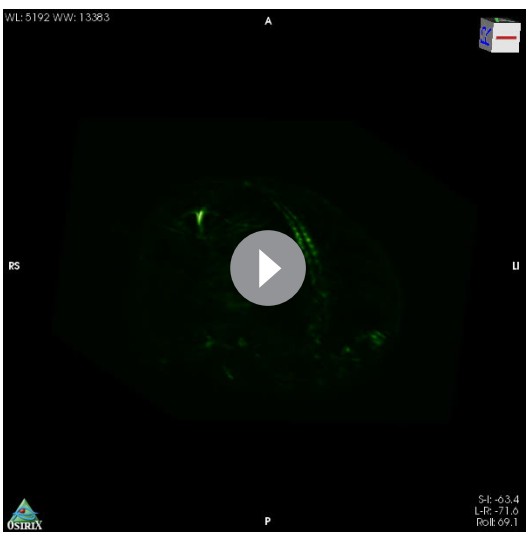

**Video 2.** 3D-rendering of dural lymphatics (green) in a 6.7-year-old common marmoset from skull-stripped subtraction T2-FLAIR images (horizontal view, 180 degrees, 7 frames/minute).
DOI: https://doi.org/10.7554/eLife.29738.015

The ability to image the meningeal lymphatics noninvasively immediately suggests the possibility of studying potential abnormalities in human neurological disorders.

## Materials and methods

### Approvals

We carried out human studies under a protocol (NCT02504840) approved by NIH Institutional Review Board. Informed consent was obtained from all participants. Formalin-fixed human brains were attained at autopsy after obtaining consent from the next of kin. Animal studies were performed under a protocol approved by the Institutional Animal Care and Use Committee.

### Human imaging

We studied five healthy volunteers (three women, age range 28–53 years) and obtained scans on a 3-tesla MRI unit (Skyra, Siemens Healthcare, Erlangen, Germany), using the body coil for radiofrequency transmission and a 32-element phased array coil for reception.

Prior to injection of gadolinium-based contrast agent, the following high-resolution MRI sequences were collected:

1. Whole-brain T1-Magnetization Prepared Rapid Acquisition of Gradient Echoes (MPRAGE, sagittal 3D turbo-fast low angle shot [TFL] sequence, acquisition matrix 256 × 256, isotropic resolution 1 mm, 176 slices, repetition time [TR]/echo time [TE]/inversion time [TI]=3000/3/900 ms, flip angle 9, acquisition time 5 min 38 s);
2. Limited T2-weighted Fluid Attenuation Inversion Recovery (FLAIR, coronal 2D acquisition over the superior sagittal sinus, field-of-view 256 × 256, 22 slices, reconstructed in-plane resolution 0.25 mm x 0.25 mm, 42 contiguous 3 mm slices, TR/TE/TI = 6500/93/2100 ms, echo train length 17, bandwidth 80 Hz/pixel, acquisition time 5 min), optimized for detection of gadolinium-based contrast agent in the subarachnoid space (*Absinta et al., 2015*);
3. Black-blood scan (coronal 2D acquisition, Sampling Perfection with Application optimized Contrasts using different flip angle Evolution [SPACE] sequence, field-of-view 174 × 174, matrix 320 × 320, reconstructed in-plane resolution 0.27 × 0.27 mm, 64 contiguous 0.5 mm sections, TR/TE = 938/22 ms, echo train length 35, bandwidth 434 Hz/pixel, acquisition time 7 min 50 s). A series of 2 or three overlapping coronal acquisitions were acquired to cover most of the cerebral hemispheres;
4. Whole-brain T2-FLAIR scan (coronal 3D acquisition, SPACE sequence, field-of-view 235 × 235, matrix 512 × 512, reconstructed in-plane resolution 0.46 × 0.46 mm, 176 1 mm sections, TR/TE/TI = 4800/354/1800 ms, nonselective inversion pulse, echo-train length 298, bandwidth 780 Hz/pixel, acceleration factor 2, acquisition time 14 min);
5. Whole-brain T1-SPACE (axial 3D acquisition, acquisition matrix 256 × 256, isotropic resolution 0.9 mm, 112 sections, TR/TE = 600/20 ms, flip angle 120, echo-train length 28, acquisition time 10 min).

We repeated these acquisitions after injection of gadobutrol (Gadavist, Bayer HealthCare, Whippany, NJ) in all five participants. In 2 of the participants, we also repeated the entire protocol prior to and following intravenous injection of gadofosveset (*Lauffer et al., 1998*) (Ablavar, Lantheus Medical Imaging, N Billerica, MA). Injections followed the manufacturer's suggested dosing (0.1 mmol/kg for gadobutrol, 0.03 mmol/kg for gadofosveset).

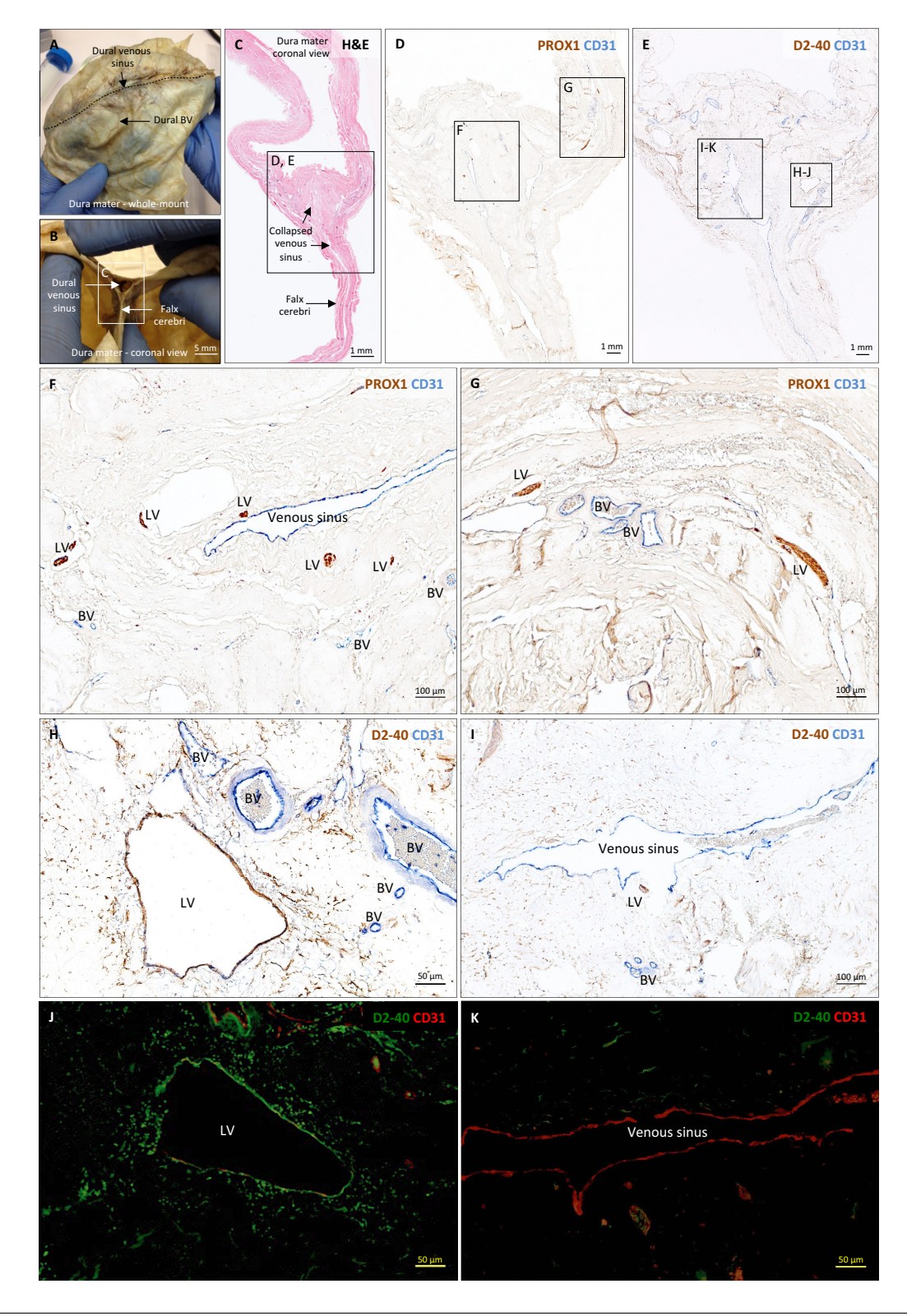

**Figure 3.** Histopathology of human dural lymphatic vessels. (A–B) Whole-mount and coronal views of the human dura mater before sampling for histological analysis. The dotted line in A shows where the superior sagittal sinus runs within the dural layers. (C) Coronal section of human dura stained with H and E to highlight anatomical features of interest, including the falx cerebri and the dural venous sinus. Note the distortion of the dura after paraffin embedding in comparison to B. (D, F, G) Within the dura mater, lymphatic and blood vessels can be differentiated using double

*Figure 3 continued on next page*

*Figure 3 continued*

immunostaining for PROX1 (a transcription factor involved in lymphangiogenesis, nuclear staining) and CD31 (a vascular endothelial cell marker). (E, H–K) Similarly, lymphatic and blood vessels can be differentiated using immunohistochemical (E, H, I) and immunofluorescent (J, K) double staining for podoplanin (D2-40, endothelial membrane staining) and CD31. Red blood cells are seen within blood vessels, but not within lymphatic vessels. Insets (F–I) were rotated relative to the original figures in D and E. *Abbreviations:* H and E: hematoxylin and eosin; LV: lymphatic vessels; BV: blood vessels.
DOI: https://doi.org/10.7554/eLife.29738.007

The following source data and figure supplements are available for figure 3:

**Source data 1.** Table of human tissue sampling.
DOI: https://doi.org/10.7554/eLife.29738.010
**Figure supplement 1.** Histopathology of human dural lymphatic vessels.
DOI: https://doi.org/10.7554/eLife.29738.008
**Figure supplement 2.** Positive and negative control staining for lymphatic vessels.
DOI: https://doi.org/10.7554/eLife.29738.009

## Marmoset imaging

We studied three healthy adult common marmosets (*Callithrix jacchus*) (one female, two males, age range 4–11 years). After the baseline MRI, experimental autoimmune encephalomyelitis (EAE) was induced in 2 marmosets with 0.2 mg of fresh-frozen human white matter homogenate as previously described (*Gaitán et al., 2014*). Marmosets were placed in a sphinx position within the magnet, and scans were obtained on a 7-tesla MRI unit (Avance AVIII, Bruker BioSpin, Billerica, MA, USA). Data acquisition was performed in transmit-only/receive-only mode using a homemade linear birdcage coil (120 mm inner diameter) as a radiofrequency transmitter and a homemade, 8-channel radio-frequency surface-array receiver coil assembly placed over the head of the animal. Prior to injection of contrast agent, we collected:

1. Whole-brain T1-weighted MPRAGE scan (coronal 3D acquisition, Modified Driven Equilibrium Fourier Transform [MDEFT] sequence, in-plane voxel size 0.15 mm x 0.15 mm, 36 contiguous 1 mm sections, TR/TE/TI = 12.5/4/1200 ms, flip angle 12 degrees, 2 segments of 1800 ms, acquisition time 7 min);
2. Whole-brain T2-FLAIR scan (coronal 2D acquisition, Rapid Acquisition with Relaxation Enhancement [RARE] sequence, voxel size 0.15 mm x 0.15 mm, 36 contiguous 1 mm sections, TR/TE/TI = 10,000/36/2500 ms, flip angle 90–180 degrees, 2 averages, acquisition time 13 min).

We performed the same scans following intravenous injection of gadobutrol and gadofosveset, in two different MRI sessions. Injections used single (0.1 mmol/kg for gadobutrol, 0.03 mmol/kg for gadofosveset) or triple the recommended human dosing (0.3 mmol/kg for gadobutrol, 0.09 mmol/kg for gadofosveset).

## Image processing

Scanner-generated DICOM images were converted into NIFTI files for postprocessing. Using MIPAV software (https://mipav.cit.nih.gov), precontrast scans were rigidly registered to postcontrast scans, and subtraction images were created for anatomical identification of dural lymphatic vessels. 3D skull-stripped subtraction images were then imported into Osirix software for maximum intensity projection (MIP) 3D rendering. The same postprocessing was performed for scans acquired after gadofosveset injection, and a direct comparison between the two gadolinium-based contrast agents was made.

## Neuropathological evaluation of human and primate dural lymphatic vessels

Neuropathological evaluation focused on human and marmoset brain dura mater samples. Multiple human dura samples were obtained from 2 formalin-fixed brains (60- and 77-year-olds with long-standing progressive multiple sclerosis) and from a 33-year-old with refractory epilepsy undergoing anterior temporal lobectomy (*Figure 3—source data 1*). Primate samples were obtained from 3 adult common marmosets (2 with EAE) [*Figure 4—source data 1*]. After general anesthesia and transcardial perfusion of 4% paraformaldehyde, marmoset brains were extracted and stored in 10%

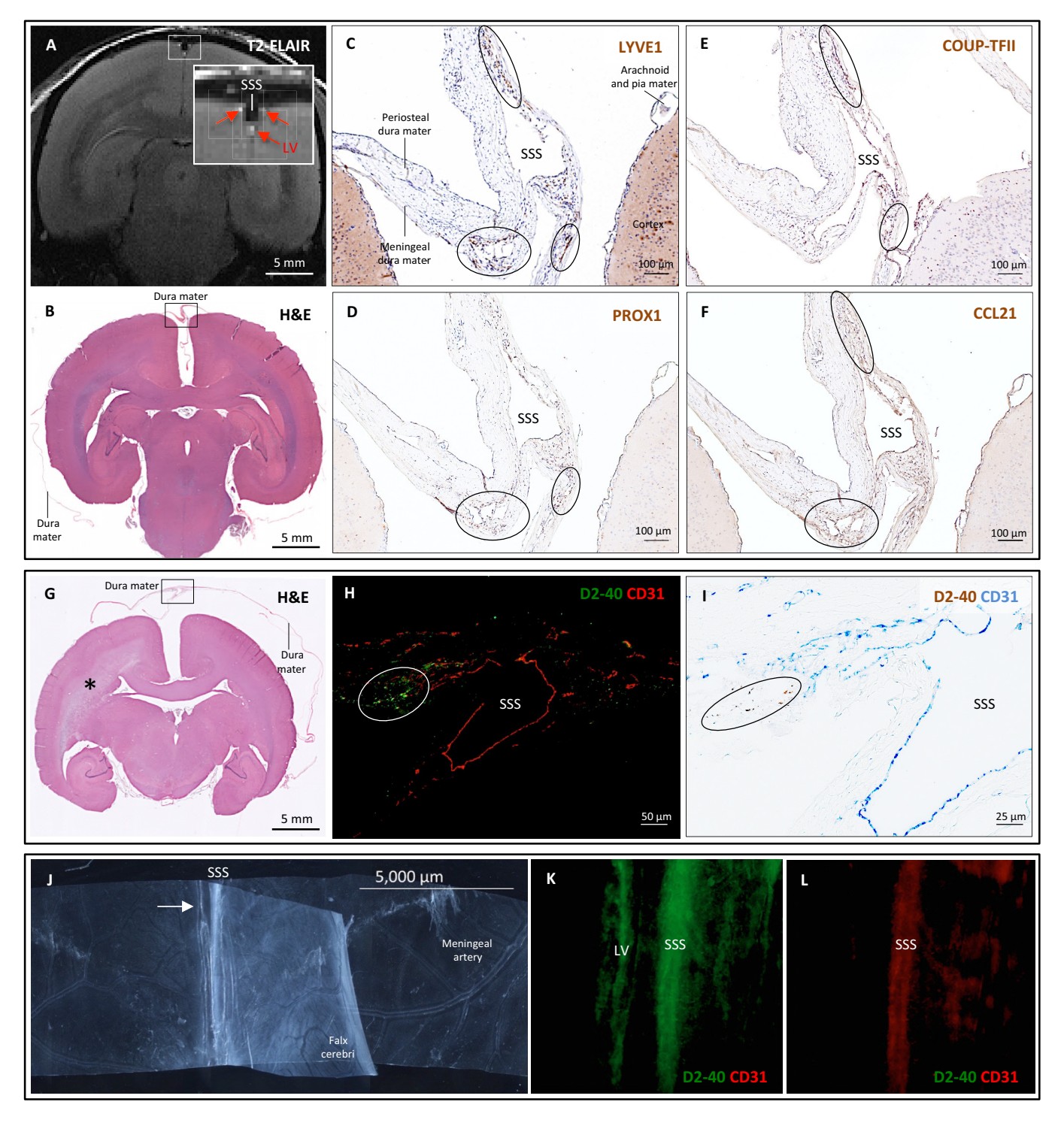

**Figure 4.** Histopathology of dural lymphatic vessels in common marmosets. MRI-histopathological correlation in a 4.4-year-old marmoset (**A–F**). (**A, B**) Postcontrast coronal T2-FLAIR showing three enhancing lymphatic vessels within the dura, and corresponding H and E section for anatomical reference (area of interest in the box). (**C-F**) Three clusters of cells (circles), surrounding the SSS and positive for lymphatic endothelial cell markers, correspond to the three enhancing areas seen on MRI (**A**). LYVE-1 is a lymphatic endothelial cell marker (membrane staining), PROX1 and COUP-TFII are transcription factors involved in lymphangiogenesis (nuclear staining), and CCL21 is a chemokine implicated in lymphatic transmigration. Higher magnifications are shown in *Figure 4—figure supplement 1*. 10.3-year-old marmoset (**G-I**). (**G**) H and E coronal section showing the brain parenchyma, meninges, and area of interest for lymphatics for anatomical reference (box). This animal did not recover after undergoing general anesthesia for blood tuberculosis

*Figure 4 continued on next page*

*Figure 4 continued*

testing, and at necropsy a stroke was identified in one hemisphere (asterisk). (H-I) Lymphatic (circles) and blood vessels were differentiated using double staining for podoplanin (D2-40, endothelial membrane staining) and vascular endothelial cell marker (CD31), respectively. 3.7-year-old marmoset (J-L). (J) Whole-mount of the marmoset dura, including the SSS. The high level of vascularization of the dura can be appreciated. The arrow indicates the area shown in K and L. (K, L) Double staining for podoplanin D2-40 for lymphatics, and CD31 for vascular endothelial cells, show the presence of a linear vascular structure parallel to the SSS, positive for podoplanin D2-40 but not CD31. The SSS is positive for both markers, probably because of antibody entrapment during immunofluorescence staining of the whole-mount dura. Abbreviations: H and E: hematoxylin and eosin; LV: lymphatic vessels; SSS: superior sagittal sinus.

DOI: https://doi.org/10.7554/eLife.29738.011

The following source data and figure supplement are available for figure 4:

**Source data 1.** Table of marmoset tissue sampling.

DOI: https://doi.org/10.7554/eLife.29738.013

**Figure supplement 1.** Histopathology of marmoset dural lymphatic vessels.

DOI: https://doi.org/10.7554/eLife.29738.012

---

formalin. For primate brains, MRI-matched histological sections were achieved via 7T MRI of the fixed brain and subsequent gross sectioning with an individualized, MRI-designed, 3D-printed cutting box, as previously described (*Absinta et al., 2014*; *Guy et al., 2016*; *Luciano et al., 2016*).

Coronal and longitudinal 3 to 7 μm paraffin sections of the dura mater were stained with hematoxylin and eosin (H and E). Immunohistochemical and immunofluorescent analysis for classical lymphatic endothelial cell markers [lymphatic vessel endothelial hyaluronan receptor 1 (LYVE-1), podoplanin (D2-40), prospero homeobox protein 1 (PROX1), COUP transcription factor 2 (COUP-TFII) and CCL21], were performed on representative slides. Selective double staining for lymphatic *vs.* vascular endothelial markers (D2-40/CD31 and PROX1/CD31) were also executed on representative sections and implemented for histological quantification; the maximum diameter was recorded for each lymphatic vessel, and descriptive statistics were computed. We also stained for lymphocytes (CD3) and monocytes/macrophages (CD68). Five-μm paraffin sections of the human skin were used as positive controls for lymphatic endothelium cell marker assessment (*Figure 3—figure supplement 2*). For comparison, 10 paraffin human brain tissue blocks, selected for the presence of nearly intact leptomeningeal architecture, were implemented to assess the presence of lymphatics within the subarachnoid space and cerebral parenchyma.

## Antibodies

Source, antibody type, and dilution are indicated sequentially as follows: LIVE-1 (Abcam, UK, rabbit polyclonal, 1:200); podoplanin D2-40 (AbD Serotec, Hercules, CA, mouse monoclonal, 1:50); PROX-1 (AngioBio, San Diego, CA, rabbit polyclonal, 1:50); COUP TFII (R&D Systems, Minneapolis, MN, mouse monoclonal, 1:200); CCL21 (Abcam, rabbit polyclonal, 1:200); CD31 (Abcam, rabbit polyclonal, 1:50); CD68 KP1 (Thermofisher, Waltham, MA, mouse monoclonal, 1:100); CD3 (Dako, Santa Clara, CA, rabbit polyclonal, 1:200); goat anti-mouse Alexafluor 488 IgG (Invitrogen, Carlsbad, CA, 1:250) and goat anti-rabbit Alexafluor 594 IgG antibodies (Invitrogen, 1:250).

## Immunohistochemistry

Immunostaining was performed on 3 to 7 μm-thick consecutive paraffin sections with antibodies to LYVE1, D2-40 podoplanin, PROX1, COUP-TFII, CCL21, CD31, CD68, CD3 and visualized with 3,3'-diaminobenzidine (DAB). Briefly, after deparaffinization, sections were rinsed in triphosphate-buffered saline (TBS) for 10 min each, processed for antigen retrieval, and treated in 3% hydrogen peroxide for 10 min. Sections were blocked with a protein-blocking solution for 20 min and incubated with the primary antibodies overnight at 4°C or 1 hr at room temperature, then rinsed and incubated with secondary antibodies for 30 min. The immunoreaction was visualized with DAB. After washing, sections were counterstained with 10% hematoxylin. Stained sections were visualized and digitized (Axio Observer Z.1, Carl Zeiss Microscopy, NY, USA). Selective double staining for lymphatic and vascular endothelium markers (D2-40 podoplanin/CD31, PROX1/CD31), as well double staining for D2-40 podoplanin/PROX1,were also executed on representative sections. Double staining was performed using, in sequence, DAB horseradish peroxidase (HRP) and Vector Blue alkaline

phosphatase (AP) methods. As negative control, sections of human skin, human dura mater, and marmoset brain were stained without the primary antibodies; no background staining and/or non-specific binding was noted (*Figure 3—figure supplement 2*).

## Immunofluorescence

After deparaffinization, sections were treated using the same method as described above. Sections were incubated with a cocktail of antibodies to D2-40 podoplanin and CD31 for 2 hr at room temperature, then rinsed and incubated with secondary antibodies with Alexafluor 488 goat anti-mouse (Invitrogen, dilution 1:250) and Alexafluor 594 goat anti rabbit IgG antibodies (Invitrogen, dilution 1:250), respectively, for 30 min at room temperature. After washing, sections were mounted with DAPI Immuno Mount. Stained sections were visualized and digitized (Axio Observer Z.1, Carl Zeiss Microscopy, NY, USA).

## Acknowledgements

The Intramural Research Program of NINDS supported this study. We thank the study participants, Irene Cortese and the NINDS Neuroimmunology Clinic for patient recruitment and care, the NIH Functional Magnetic Resonance Imaging Facility for MRI support, Afonso C Silva and his laboratory for access to the 7T animal scanner and customized hardware, Steven Jacobson and Emily Leibovitch for collaboration on our animal studies, and Roger Depaz for assistance in animal preparation.

## Additional information

### Competing interests

Martina Absinta: Dr. Absinta was partially supported by a National Multiple Sclerosis Society (NMSS) fellowship award #FG 2093-A-1 and holds a Marilyn Hilton Award for Innovation in MS research from the Conrad N. Hilton Foundation. Daniel S Reich: Dr. Reich received research support from collaborations with the Myelin Repair Foundation and Vertex Pharmaceuticals, unrelated to the present study. The other authors declare that no competing interests exist.

### Funding

| Funder | Grant reference number | Author |
| --- | --- | --- |
| National Institute of Neurological Disorders and Stroke | Intramural Research Program | Martina Absinta<br>Seung-Kwon Ha<br>Govind Nair<br>Pascal Sati<br>Nicholas J Luciano<br>Kareem A Zaghloul<br>Daniel S Reich |
| National Multiple Sclerosis Society | Postdoctoral Fellowship | Martina Absinta |
| Conrad N. Hilton Foundation | Marilyn Hilton Award for Innovation in Multiple Sclerosis Research | Martina Absinta |
| National Cancer Institute | Intramural Research Program | Maryknoll Palisoc<br>Stefania Pittaluga |
| National Institute on Aging | R01AG034113 | Antoine Louveau<br>Jonathan Kipnis |
| Lymphatic Education and Research Network | Postdoctoral fellowship | Antoine Louveau |

The funders had no role in study design, data collection and interpretation, or the decision to submit the work for publication.

## Author contributions

Martina Absinta, Conceptualization, Data curation, Formal analysis, Methodology, Writing—original draft, Writing—review and editing; Seung-Kwon Ha, Govind Nair, Pascal Sati, Antoine Louveau, Stefania Pittaluga, Data curation, Formal analysis, Writing—review and editing; Nicholas J Luciano, Kareem A Zaghloul, Data curation, Writing—review and editing; Maryknoll Palisoc, Formal analysis, Methodology; Jonathan Kipnis, Conceptualization, Writing—review and editing; Daniel S Reich, Conceptualization, Data curation, Formal analysis, Supervision, Funding acquisition, Investigation, Methodology, Writing—original draft, Project administration, Writing—review and editing

## Author ORCIDs

Martina Absinta [ID] https://orcid.org/0000-0003-0276-383X
Seung-Kwon Ha [ID] http://orcid.org/0000-0003-3671-4454
Govind Nair [ID] http://orcid.org/0000-0003-3725-615X
Kareem A Zaghloul [ID] https://orcid.org/0000-0001-8575-3578
Daniel S Reich [ID] https://orcid.org/0000-0002-2628-4334

## Ethics

Human subjects: We carried out human studies under a protocol (NCT02504840) approved by the NIH Institutional Review Board. Informed consent was obtained from all participants.
Animal experimentation: Animal studies were performed under a protocol approved by the Institutional Animal Care and Use Committee of the NIH (NIH Animal Studies Protocol 1308-15).

## Decision letter and Author response

Decision letter https://doi.org/10.7554/eLife.29738.017
Author response https://doi.org/10.7554/eLife.29738.018

## Additional files

### Supplementary files

• Transparent reporting form
DOI: https://doi.org/10.7554/eLife.29738.016

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
