## [Decision Letter]

Thank you for submitting your article "The primate meninges harbor lymphatic vessels that can be visualized noninvasively by MRI" for consideration by *eLife*. Your article has been reviewed by three peer reviewers, and the evaluation has been overseen by a Reviewing Editor and David Van Essen as the Senior Editor.

The reviewers have discussed the reviews with one another and the Reviewing Editor has drafted this decision to help you prepare a revised submission.

Summary:

This report is a follow-up on the recent publications on the meningeal lymphatic vessels in rodents. They study image human and nonhuman primates using T2-FLAIR and T1-weighted imaging in combination with systemic administration of contrast agents. The analysis shows that the meningeal lymphatic vessels in human and nonhuman primate meninges are very similar to those described in rodents.

Essential revisions:

1) In this study, and in the previous work on the glymphatic system, there is little or no mention of the well-known Virchow-Robin perivascular spaces, easily observable using high resolution MRI (e.g. Thomas et al., MRM 2004). Because these features of brain anatomy are well known, having been studied for over a century, and they are also observable with MRI, it is quite surprising that they were not mentioned in the present manuscript. Were they not observed at all? If indeed they comprise a part of the lymphatic circulation, one would expect them to appear with significant contrast in the gadobutrol labelled images. The authors could provide additional visualisation.

2) Lymphatic flow. One of the criticisms of the 'glymphatic system' concept concerns the shortage of evidence that the lymph has a directional flow within it. A study of the kind described in this manuscript could help to rectify this lack – if a time series of images were obtained, displaying successive enhancement of lymphatic channels in the order predicted by the theory. The authors might want to comment on this possibility.

3) The MRI imaging is based on the use of two gadolinium-based tracers, one (gadofosveset) which is confined to the blood by binding to serum albumin, and one (gadobutrol) which can exit through blood vessel walls and pass into the lymph vessels. The latter showed accumulation in (putative) lymph vessels in the dura, while the former did not. This is encouraging but there are the following problems:

a) The putative lymphatics are barely visible, especially in the subtraction images in Figure 2. The authors should increase the intensity and the contrast of the images (taking care to specify what manipulations have been done in the figure legend) – for example, pasting Figure 2's subtraction images into Powerpoint and increasing both the intensity and contrast by 40% leads to much more easily visible lymphatics. A similar process should be applied to Figure 2—figure supplement 1 for the subtraction images, if the authors wish to convince the reader that there are no lymphatics visible with gadofosveset.

b) The videos show the extent of the lymphatics far more impressively than do the coronal sections in Figure 1 and 2. Panels should be added to Figure 1 and Figure 2 showing sagittal sections or simply a picture of the 3D rendering in the videos. The contrast should also be increased in the videos.

4) Ultimately the MRI data need validating by immunocytochemistry with lymphatic specific markers, and this is attempted in Figure 3 and 4. Again there are problems:

a) Figure 3 is very poorly presented for readers not familiar with the detailed anatomy. I cannot understand what Figure 3 shows; add some explanatory labels. For panels D-E we need some clue as to exactly where in the brain the tissue is located – show a picture like that in Figure 4, with rectangles to approximately locate the images presented in D-E. No scale bar size is stated for panels D and E and K.

b) In humans, Prox 1 or podoplanin (brown) are used as a lymphatic marker and CD31 (blue) as an endothelial marker. Unfortunately the images in panels 3D-E are so faint that colors are not visible (in contrast the immunofluorescence in panels J-K is much clearer). In the magnified panels colors are visible but the background labeling is high. Any dark brown label is attributed to the presence of lymphatics (e.g. there are 5 structures annotated LV in panel F) yet it is unclear how these vessels relate to those seen in the MRI pictures, where there are 3 lymph vessels in Figure 1. Were the IHC and the MRI done on the same brains? If so, is it not possible to state whether the number of lymph vessels found by MRI is the same as is found by IHC? The fact that the IHC finds more vessels (below the resolution of MRI?) at least needs to be discussed. Also, the fact that podoplanin is in cells other than lymph vessels should be mentioned (e.g. kidney podocytes and possibly capillary pericytes, which could confound the interpretation: https://www.ncbi.nlm.nih.gov/pubmed/25908104).

c) For the marmoset data, the text states that the number of lymphatics seen with MRI was the same as with IHC, yet the number of lymphatics seen in Figure 4 could be 2 or 3 (enhance the intensity and contrast, and label them with red arrows), while in Figure 4 It is unclear exactly which brownish structures are being defined as lymphatics – they need explaining better, e.g. by drawing a dashed red line round them.

d) For both human and marmoset tissue, it is essential to show the (hopefully absent) labeling when the primary antibody for PROX1 or D2-40 is omitted from the IHC. Ideally this should be in brain, but skin would be an acceptable substitute if brain is unfeasible.

e) Figure 3—figure supplement 2 and its associated text is fairly peripheral to the main point of this paper – it might be better in a separate paper combined with more rigorous data showing lymphatic trafficking of immune cells.

---

## [Author Response]

Essential revisions:1) In this study, and in the previous work on the glymphatic system, there is little or no mention of the well-known Virchow-Robin perivascular spaces, easily observable using high resolution MRI (e.g. Thomas et al., MRM 2004). Because these features of brain anatomy are well known, having been studied for over a century, and they are also observable with MRI, it is quite surprising that they were not mentioned in the present manuscript. Were they not observed at all? If indeed they comprise a part of the lymphatic circulation, one would expect them to appear with significant contrast in the gadobutrol labelled images. The authors could provide additional visualisation.

We thank the reviewers for raising this point. We were able to image the meningeal lymphatic vessels because there is no blood-meningeal barrier in the dura mater; hence, gadolinium contrast can quickly extravasate into the connective tissue, from where it collects into the lymphatic vessels. On the other hand, the Virchow-Robin perivascular spaces in the brain parenchyma, which (as the reviewer correctly points out) are commonly seen on conventional clinical MRI scans, are known not to enhance after intravenously injected gadolinium contrast. This is most likely related to the presence of an intact blood-brain barrier which, under physiological conditions, is not permeable to contrast agents. In this context, we note that our approach does not address the recently described “glymphatic” system, which may comprise the Virchow-Robin spaces, the noninvasive imaging of which would require an altogether different approach from what we have presented in this paper. These points have been better clarified in the manuscript (1^st^ paragraph of the Results/Discussion section).

2) Lymphatic flow. One of the criticisms of the 'glymphatic system' concept concerns the shortage of evidence that the lymph has a directional flow within it. A study of the kind described in this manuscript could help to rectify this lack – if a time series of images were obtained, displaying successive enhancement of lymphatic channels in the order predicted by the theory. The authors might want to comment on this possibility.

We thank the reviewers for this comment as well. As mentioned in the reply to point 1, imaging the glymphatic system would require a different approach from the one we have presented. Indeed, even though we collected time series images throughout the first hour after injection of contrast agent (an example of which is shown in Figure 2—figure supplement 1), we did not observe clear dynamic changes (e.g. flow) in the imaging characteristics of the meningeal vessels. More subtle dynamic changes might have been observed had we done more frequent temporal sampling, however the requirement for extremely high spatial resolution to visualize the meningeal lymphatic vessels made this impossible. We have now included a comment in the Discussion.

3) The MRI imaging is based on the use of two gadolinium-based tracers, one (gadofosveset) which is confined to the blood by binding to serum albumin, and one (gadobutrol) which can exit through blood vessel walls and pass into the lymph vessels. The latter showed accumulation in (putative) lymph vessels in the dura, while the former did not. This is encouraging but there are the following problems:a) The putative lymphatics are barely visible, especially in the subtraction images in Figure 2. The authors should increase the intensity and the contrast of the images (taking care to specify what manipulations have been done in the figure legend) – for example, pasting Figure 2's subtraction images into Powerpoint and increasing both the intensity and contrast by 40% leads to much more easily visible lymphatics. A similar process should be applied to Figure 2—figure supplement 1 for the subtraction images, if the authors wish to convince the reader that there are no lymphatics visible with gadofosveset.

We thank the reviewers for this recommendation. The figures have been modified accordingly.

b) The videos show the extent of the lymphatics far more impressively than do the coronal sections in Figure 1 and 2. Panels should be added to Figure 1 and Figure 2 showing sagittal sections or simply a picture of the 3D rendering in the videos. The contrast should also be increased in the videos.

We added a new figure (Figure 1—figure supplement 1) including static snapshots of the 3D rendering of the putative human dural lymphatics and anatomical labels for ease of interpretation, as suggested by the reviewers.

4) Ultimately the MRI data need validating by immunocytochemistry with lymphatic specific markers, and this is attempted in Figure 3 and 4. Again there are problems:a) Figure 3 is very poorly presented for readers not familiar with the detailed anatomy. I cannot understand what Figure 3 shows; add some explanatory labels. For panels D-E we need some clue as to exactly where in the brain the tissue is located – show a picture like that in Figure 4, with rectangles to approximately locate the images presented in D-E. No scale bar size is stated for panels D and E and K.

We thank the reviewers for this recommendation and apologize that the first version was difficult to decipher. These figures have been modified accordingly.

b) In humans, Prox 1 or podoplanin (brown) are used as a lymphatic marker and CD31 (blue) as an endothelial marker. Unfortunately the images in panels 3D-E are so faint that colors are not visible (in contrast the immunofluorescence in panels J-K is much clearer). In the magnified panels colors are visible but the background labeling is high. Any dark brown label is attributed to the presence of lymphatics (e.g. there are 5 structures annotated LV in panel F) yet it is unclear how these vessels relate to those seen in the MRI pictures, where there are 3 lymph vessels in Figure 1. Were the IHC and the MRI done on the same brains? If so, is it not possible to state whether the number of lymph vessels found by MRI is the same as is found by IHC? The fact that the IHC finds more vessels (below the resolution of MRI?) at least needs to be discussed. Also, the fact that podoplanin is in cells other than lymph vessels should be mentioned (e.g. kidney podocytes and possibly capillary pericytes, which could confound the interpretation: https://www.ncbi.nlm.nih.gov/pubmed/25908104).

The Discussion (Results/Discussion 3^rd^ paragraph) addresses differences in the number of lymphatics seen on MRI vs. histopathology, as follows: “It is worth noting that the lymphatics visualized by MRI are large slow-flow lymphatic ducts, whereas blind-ending and small lymphatic capillaries, clearly seen by histopathology (Figure 3 and Figure 3—figure supplement 1), are unlikely to be revealed by MRI.”

We agree that lymphatic immunohistochemical markers are not totally specific for lymphatic vessels (please see Results/Discussion 4^th^ paragraph), which is the reason we needed to test several of them (LYVE-1, podoplanin D2-40, PROX1, COUP-TFII, and CCL21) and to compare results on consecutive sections of dura mater. The suggested reference has been now added to the manuscript.

For the humans, the imaging and histopathology have been performed, of necessity, in different people (please see Figure 3—source data 1 for human tissue sampling), as described in the Materials and methods section. On the animal side, we studied a total of 5 marmosets for this study: 2 of which were used exclusively for MRI-histopathological correlations (please see Figure 4—source data 1), 2 for gabobutrol vs. gadofosvet MRI in vivo experiments, and 1 for both the gabobutrol vs. gadofosvet MRI in vivo experiments and the histopathological characterization.

The relevant figure and legends have been modified according to the request of the reviewers.

c) For the marmoset data, the text states that the number of lymphatics seen with MRI was the same as with IHC, yet the number of lymphatics seen in Figure 4 could be 2 or 3 (enhance the intensity and contrast, and label them with red arrows), while in Figure 4 It is unclear exactly which brownish structures are being defined as lymphatics – they need explaining better, e.g. by drawing a dashed red line round them.

We apologize for any confusion. For the marmoset data, we stated in the text that the 3 areas enhancing on MRI correspond to clusters of cells positive for lymphatic markers (4^th^ paragraph of the Results/Discussion section). Unfortunately, it was not always straightforward to recognize the vessel-like structures, likely due to the fragility of the dura mater after preparation and immunohistochemical processing. The figure and the legend have been modified according to the request of the reviewers.

d) For both human and marmoset tissue, it is essential to show the (hopefully absent) labeling when the primary antibody for PROX1 or D2-40 is omitted from the IHC. Ideally this should be in brain, but skin would be an acceptable substitute if brain is unfeasible.

We have performed negative control staining in human dura and skin as well as in marmoset brain. We included them in Figure 3—figure supplement 3. They show no background staining and/or nonspecific binding. An additional sentence has been added to the Materials and methods section.

e) Figure 3—figure supplement 2 and its associated text is fairly peripheral to the main point of this paper – it might be better in a separate paper combined with more rigorous data showing lymphatic trafficking of immune cells.

The figure has been deleted accordingly.